# Preparation of Stable Phase Change Material Emulsions for Thermal Energy Storage and Thermal Management Applications: A Review

**DOI:** 10.3390/ma15010121

**Published:** 2021-12-24

**Authors:** Liu Liu, Jianlei Niu, Jian-Yong Wu

**Affiliations:** 1Department of Applied Biology & Chemical Technology, The Hong Kong Polytechnic University, Hung Hom, Kowloon, Hong Kong; liu66.liu@connect.polyu.hk; 2Department of Building Environment and Energy Engineering, The Hong Kong Polytechnic University, Hung Hom, Kowloon, Hong Kong; jian-lei.niu@polyu.edu.hk

**Keywords:** phase change material, emulsions, paraffin, thermophysical properties, thermal energy storage, thermal management

## Abstract

Thermal energy storage (TES) is an important means for the conservation and efficient utilization of excessive and renewable energy. With a much higher thermal storage capacity, latent heat storage (LHS) may be more efficient than sensible heat storage. Phase change materials (PCMs) are the essential storage media for LHS. PCM emulsions have been developed for LHS in flow systems, which act as both heat transfer and thermal storage media with enhanced heat transfer, low pumping power, and high thermal storage capacity. However, two major barriers to the application of PCM emulsions are their instability and high degree of supercooling. To overcome these, various strategies have been attempted, such as the reduction of emulsion droplet size, addition of nucleating agents, and optimization of the formulation. To the best of our knowledge, however, there is still a lack of review articles on fabrication methods for PCM emulsions or their latest applications. This review was to provide an up-to-date and comprehensive summary on the effective strategies and the underlying mechanisms for the preparation of stable PCM emulsions and reduction of supercooling, especially with the organic PCMs of paraffin. It was also to share our insightful perspectives on further development and potential applications of PCM emulsions for efficient energy storage.

## 1. Introduction

Global warming is currently recognized as one of the most alarming and threatening problems to human life on earth. A major cause for global warming is attributed to the rapid consumption of carbon energy generated by burning fossil fuels. Therefore, wide attention has been paid to the development of measures for conservation and efficient utilization of energy. Although some alternative and sustainable energy sources have been sought and increasingly used such as solar and wind energy, thermal energy is still the most important and widely used in many fields and circumstances. Thermal energy storage (TES) is a useful measure for the harvest and conservation of excessive energy and for effective utilization where and when it is most needed. Among various TES technologies, the storage of latent heat is regarded as a highly efficient strategy because of the much higher energy capacity than sensible heat. Phase change materials (PCMs) are the indispensable media for latent heat storage (LHS). For LHS in fluid media, considerable research effort has been devoted to the development of PCM-based heat transfer fluids (HTFs) in the forms of slurries and emulsions. It may bring out several benefits such as higher overall heat transfer coefficient during the phase change process of PCM [1,2], higher thermal storage capacity, less pumping power consumption under high energy demand [3], and simple and diverse system design, in comparison to conventional HTF (e.g., water) or stationary LHS units with solid–liquid PCMs.

As the most extensively studied category [4,5,6,7], microencapsulated PCM (MPCM) slurries are formed by simply dispersing MPCM solid particles into water. However, the complex synthesis route and shell breakup risk of MPCM, and gravity-driven phase separation of MPCM and water, due to its relatively large particle size (usually 1–100 μm), are unfavorable for practical applications. Therefore, PCM emulsions are formed by dispersing PCM as fine droplets into an immiscible carrier fluid with the assistance of surfactants. With the merits of shell-free, smaller droplet size and easy production, PCM emulsions are recognized as the promising PCM-based HTF in various fields.

Many previous studies have reported on thermal storage properties of PCM emulsions, including emulsion stability, thermal properties, and rheological behaviors with respect to the selection and content of PCMs and surfactants and the preparation methods, which have been summarized in a few review articles (Table 1). Delgado et al. [8] reviewed the physical properties and heat transfer performance of PCM emulsions and MPCM slurries. A later review by Youssef et al. [9] summarized the development of PCM slurries in a wider range, including clathrate hydrate slurries (CHS). Shao et al. [10] published a more comprehensive review on PCM emulsions covering the fabrication methods, physical properties, heat transfer performances and HAVC applications. More recently, Wang et al. [11] and O’Neill et al. [12] reviewed the studies on the stability, supercooling, thermal conductivity, rheological behavior, pumping power and heat transfer characteristics of PCM emulsions.

Although the thermophysical properties and heat transfer characteristics of PCM emulsions have been extensively discussed, especially in two recent review articles, the relevant applications are not covered. It is of interest to summarize the latest case studies on PCM emulsions. On the other hand, the preparation or fabrication methods are rarely reviewed comprehensively in the previous reviews, which are important for the future large-scale production and application of PCM emulsions.

Therefore, the present review is to offer an up-to-date and comprehensive summary on the various methods and the underlying mechanisms for the preparation of stable PCM emulsions with a lower degree of supercooling. The studies on the formulation of PCM emulsions were mostly collected from Elsevier’s Scopus database and Google scholar since 2010, using several keywords such as PCM emulsions, paraffin emulsions, and nano-emulsions, while the relevant application studies were summarized since 2015. The review will provide a useful reference not only for effective preparation of the PCM emulsions but also for their large-scale applications in TES systems and other related areas.

## 2. Major Characteristics of PCM Emulsions

### 2.1. Classification of PCMs

PCMs can be classified in terms of the phase change state, chemical composition or melting temperature range [13,14,15]. According to the state of phase change, PCMs can be divided into solid–solid, solid–liquid, solid–gas, and liquid–gas. Among these four types of PCMs, solid–gas and liquid–gas PCMs exhibit the highest energy storage density, but the wide volume and pressure variation during the phase change process are undesirable for their practical applications. Moreover, the solid–solid PCMs also have a low energy storage density and high phase change temperature. In comparison, the solid–liquid PCMs are more promising candidates with more favorable phase change temperatures and higher latent heats. The solid–liquid PCMs can be further divided into three categories according to their chemical composition, i.e., organic, inorganic, and eutectic mixtures (Figure 1), which have different ranges of melting temperature (T_m_). In the high-temperature range (T_m_ > 200 °C are the molten inorganic salts, including nitrates, carbonates, and chlorides which are mainly applied for industrial waste heat recovery [16] and concentrated solar power (CSP) [17]. They have the advantages of relatively high energy storage density (200–1000 J/g) and high thermal conductivity (0.4–0.7 W/m·K) but the main drawback is the corrosive effect on the storage tanks and delivery pipes [18,19]. For the medium temperature range (100–200 °C), the organic sugar alcohols and inorganic salt hydrates are most applicable but liable to phase segregation and severe supercooling up to 90 °C for pure erythritol [20]. In the low-temperature range (T_m_ < 100 °C), organic PCMs such as fatty acids and paraffin are most favorable, especially paraffin with the basic formula of CH_3_-(CH_2_)_n_-CH_3_ due to several advantages, including a broad melting temperature range (−12 °C to 135 °C), high latent heat (up to 269 J/g), low supercooling or phase segregation, high durability, and low material cost, though a major drawback is their low thermal conductivity in the range of 0.15–0.30 W/m·K, resulting in lower heat transfer rate and less effectiveness in practical applications [21]. In the following parts of the review, paraffin-based PCMs will be mainly taken as the models for the optimal formation of stable and high-performance PCM emulsions and the promising strategies for overcoming the instability and supercooling problems.

### 2.2. Classification of PCM Emulsions

Emulsion is generally a liquid dispersion consisting of two or more immiscible liquids. Water and oil are two common liquids that can form various forms of emulsions, including binary water-in-oil (W/O) or oil-in-water (O/W) emulsions and multiple oil-water-oil (O/W/O) and water-oil-water (W/O/W) (Figure 2). The O/W emulsion is the most common form of PCM emulsions. Apart from oil and water, an emulsifier is needed to form a stable emulsion. The emulsifier is usually a surface-active agent which reduces the interfacial tension (*γ*), lowering the free energy (Δ*G*), as given by [22].
(1)$$\Delta G=\gamma \frac{3V}{R}$$
where *V* is the total volume of the dispersed oil and *R* is the radius of PCM droplet, there are various categories of emulsifiers, such as phospholipids, proteins, polysaccharides and small molecule surfactants [23]. In addition to the molecular emulsifiers, nanoparticles have been used to form Pickering emulsions [24].

For PCM emulsions, the most common emulsifiers are nonionic and ionic small molecule surfactants, which are characterized by a polar hydrophilic head group and a nonpolar hydrophobic alkyl tail. The formation of the thin protective layer at the interface between water and oil could effectively prevent droplet aggregation. The activity of such surfactants can be simply quantified by calculating their hydrophile-lipophile balance (HLB) values [25]. In general, hydrophilic surfactants with relatively high HLB values (8–18) favor the formation of O/W emulsions, while hydrophobic surfactants with HLB values at 4–6 are suitable for W/O emulsions. The free energy needed for the formation of PCM emulsions is derived from mechanical forces with the high-energy methods or physicochemical processes with the low-energy methods.

Emulsions can also be divided into conventional macro-emulsions (d > 200 nm), nano-emulsions (20–200 nm), and micro-emulsions (5–50 nm) according to the droplet size range (Table 2) [23,26]. Macro-emulsions have a milky white appearance because of the relatively large droplet size, which are prone to break down through coalescence and flocculation [27,28], resulting in the formation of even larger droplets, creaming on the top layer, and eventually phase separation. Therefore, they are both kinetically and thermodynamically unstable.

Nano-emulsions, also called mini-emulsions, have smaller droplet sizes than the macro-emulsions, exhibiting a bluish transparent (below 50 nm), translucent, or creamy optical appearance. The small droplet size can effectively overcome gravity effect due to Brownian motion [29] and dramatically prolong the stable period, which means they are kinetically stable. However, nano-emulsions are non-equilibrium colloidal dispersions. Because of the extremely large interfacial area between water and oil, the system is not in the lowest thermodynamic energy state. As a result, the nano-droplets can grow by the mechanism of Ostwald Ripening [30,31], leading to a gradual reduction of interfacial area, and phase separation happens in a longer time period to reach the minimum energy state, i.e., thermodynamically unstable characteristic.

Micro-emulsions are similar to nano-emulsions in the droplet size but are thermodynamically stable. In an O/W emulsion, surfactant micelles can form when the surfactant concentration is higher than its critical micelle concentration (CMC) value. The micelles could be spherical or cylindrical, depending on multiple factors such as the type and content of surfactants and temperature. The micro-emulsion system is thermodynamically stable in the suitable temperature range. A micro-emulsion is usually formed as a dispersed oil filling into the cavity of surfactant micelles at a high surfactant concentration, e.g., 20% or above, while the oil content is relatively low [32]. As the oil content increases, the surfactant micelles are gradually enlarged and eventually disbanded by large oil droplets, and turned into nano-emulsions. Although micro-emulsions are the most stable, they are not suitable for TES systems due to their high sensitivity to changes in temperature and composition. In order to overcome instability issue, PCM nano-emulsions are regarded as the most promising selection due to their relatively high kinetic stability and robustness in practical applications.

### 2.3. Supercooling in PCM Emulsions

Supercooling, which is defined as the difference between the melting and freezing point of a substance, is unfavorable since it causes delay in the phase change and lower system efficiency. Supercooling occurring in the PCM emulsions is mainly attributed to the shift from heterogeneous to homogeneous nucleation [33], which might be more serious in nano-emulsions since a much smaller droplet size could further reduce the homogeneous nucleation rate. Based on this mechanism, the addition of nucleating agents has been widely explored as a direct measure for overcoming supercooling. However, the supercooling issue in PCM emulsions is still far from being well resolved because of many other factors than nucleation [34]. For example, the crystallization characteristics of alkanes in bulk or within micro/nanoconfinement are diverse due to the variation of their metastable or rotator phases [35,36]. Therefore, the supercooling issue is still one of the major challenges in the development of high-performance PCM emulsions.

## 3. Preparation of PCM Emulsions by High-Energy Methods

According to Equation (1), higher free energy is required for the formation of emulsions with smaller droplet sizes. With the high-energy methods, the mechanical energy increases sharply with the droplet size reduction. For example, the low-energy shaking with hand is sufficient to produce oil in water emulsions with a droplet size of 40–100 μm, but a huge amount of energy approximately in the order of 10^8^–10^10^ W/kg [37] would be required to produce nano-sized droplets.

There are several common high-energy methods, as illustrated in Figure 3, high-pressure homogenization and microfluidization, membrane emulsification, rotor-stator homogenization, and ultrasonic emulsification. Based on the literature survey, rotor-stator homogenization and ultrasonic emulsification have been most commonly applied for PCM emulsions, while high-pressure emulsification and membrane emulsification methods have rarely been applied. In fact, these emulsification techniques have been studied extensively and used widely in the food and pharmaceutical industries for several decades. As an emerging research area, PCM emulsions have been currently formulated by less diverse fabrication methods. The low cost and convenience of rotor-stator homogenizers and probe-type ultrasonic processors are attributable to their wider use in laboratory studies. The following sections describe the general working principles, then summarize the related studies.

### 3.1. General Principles

#### 3.1.1. High-Pressure Techniques

A high-pressure homogenizer (HPH) consists of a high-pressure pump and a homogenization valve with a narrow gap (2–30 μm) [43,44,45] (Figure 3A). The homogenization pressure is normally in the range of 50 and 400 MPa, with the low-pressure range for the standard homogenizers and the high-pressure range for the ultra-HPH (UHPH), respectively. Droplet size reduction occurs in the homogenization valve, whose geometry mainly determines the homogenization pressure and the characteristics of emulsions. There are three major types of valve geometry, counter jet, radial diffuser, and axial flow valves. The counter jet valve is constructed with two opposing bores, in which the two separated fluid streams are accelerated at a high velocity and then frontally impinged [44]. The radial diffuser valve consists of a mobile valve seat and an axial valve face [46], in which the upstream fluid first collides with the axial valve face and then splits radially through the gap slit between the mobile valve seat and the axial valve face. By controlling the slit width and velocity of the upstream fluid, the homogenization pressure can be adjusted to up to 150 MPa. The axial flow valve is designed with a smaller inlet diameter, where the pressurized fluid enters the valves axially with increasing velocity and collides with a needle and an annular valve seat, in which up to 400 MPa pressure can be achieved by adjusting the slit of the needle and valve seat.

Theoretically, droplet size reduction by HPH is achieved with the combined effects of shearing, turbulence, and cavitation [43]. The shearing effect plays a major role in droplet disruption, as described by the Weber number (*We* = *ρv^2^l/γ*) [47] in a laminar flow. The surface tension (*γ*) is affected by temperature and pressure (Equation (2)):(2)$${\left(\frac{\partial \gamma }{\partial P}\right)}_{T,A}={\left(\frac{\partial V}{\partial A}\right)}_{P,T}$$
where *P* is pressure, *T* temperature, *V* and *A* the volume and total surface area of the system, respectively. The turbulence is increased by the dramatic fluid velocity increment with the reducing tubing size, which is the principal cause of mixing and homogenization [44]. The corresponding droplet break-up mechanism will be introduced in the later section on rotor-stator homogenizers. Cavitation occurs in the flow regions of high velocity and low pressure, where the vapor bubbles or cavities form in the liquid and subsequently collapse [48].

The working principle of a high-pressure microfluidizer is slightly different from that of HPH. The valve or nozzle in HPH is replaced by an interaction chamber, and a pneumatic pump is used to provide high pressure of up to 40,000 psi or 275 MPa [49,50]. Homogenization occurs in the interaction chamber, which has a similar function to the counter jet valve but a higher efficiency. The application of two opposite micro-streams causes a high shear rate (up to 10^7^ s^−1^), turbulent mixing impact, and cavitation to yield fine emulsions of very small droplets. The interaction chamber has two common geometries, Y-type and Z-type. With single to multiple channels, the production level could be scaled up (Figure 4). However, microfluidizers are difficult to clean and liable to blockage.

#### 3.1.2. Membrane Emulsification

Membrane emulsification (ME) requires low energy to form fine emulsion based on a drop-by-drop mechanism (Figure 3C). The dispersed phase, either a pure liquid or coarse emulsion, is pumped under gas pressure through a porous membrane into the continuous phase, which is known as the direct or premix ME process [40,52]. It is noteworthy that the porous membrane is wetted only in the continuous phase but not in the dispersed phase. Hydrophilic membranes are used to produce O/W emulsions and hydrophobic membranes produce W/O emulsions. The transmembrane pressure (Δ*P_tm_*) is defined by [40],
(3)$$\Delta P_{tm}=P_{d}-\left(\frac{P_{c,in}-P_{c,out}}{2}\right)$$
where *P_d_* is the pressure of dispersed phase, *P_c,in_* and *P_c,out_* are the pressure of inlet and outlet of the continuous phase, respectively. The critical transmembrane pressure (*P_c_*), which is responsible for the detachment of dispersed phase droplets from the membrane surface, can be estimated by [40],
(4)$$P_{c}=\frac{4\gamma \cos\theta }{{\bar{d}}_{p}}$$
where *γ* is the interfacial tension, *θ* is the contact angle of the dispersed phase droplet and membrane surface, ${\bar{d}}_{p}$ is the average pore size of the membrane. The required pressure is much lower than that of HPH or microfluidizer, thereby the energy consumption is reduced by around 2 orders. Apart from the physicochemical properties of the dispersed and continuous phases, the operating conditions and membrane parameters can have a significant influence on the properties of emulsion.

The ME can be operated by moving either the continuous phase or the membrane [52]. The continuous phase can be in a pulsed or continuous flow, while the membrane movement can be in rotation or vibration mode. The membrane properties, including pore size, morphology, porosity, and wettability are also crucial for ME. Several hydrophilic membranes, including Shirasu porous glass (SPG) membrane [53], macroporous silica glass membrane [54], polytetrafluoroethylene (PTFE) membrane [55], and ceramic aluminum oxide (α-Al_2_O_3_) membrane [40], have been developed to fabricate O/W emulsions. Although the emulsion droplets prepared by ME can have a narrow distribution, the droplet sizes are mostly in the microscale. The synthesis of membranes with evenly distributed nanopores is still a bottleneck for ME.

#### 3.1.3. Rotor-Stator Homogenizer

The rotor-stator homogenizer is a high-shearing homogenization (HSP) device, which has a rotating metal shaft (the rotor) inside a stationary casing (the stator) (Figure 3D). It is widely used as a pre-mixing device working at a low rotation speed to produce coarse emulsions or operated at a high rotation speed for the emulsification of dispersed liquids with medium to high viscosities [56]. When the rotor starts rotating, the low pressure generated circulates the liquid from the rotor bottom in and through the stator holes out.

It is widely recognized that the droplet breakup mechanism is associated with the high shear stress in the gap between rotor and stator, which is generated by a high rotor velocity [56,57,58]. Therefore, variations in the thickness of gaps are commonly in the millimeter range, and the rotation speed varies from 1000 rpm to 30,000 rpm. The rotor is usually blade or tooth design [59], and its diameter affects the droplet size distribution. On the other hand, a recent investigation [58] has pointed out that the vortex-jet and recirculation breakup mechanism also play a distinct role at different rotation speeds. The former refers to a combination of vortex generated by the rotor and fluid jet produced in the stator holes, while the latter can be described as droplets getting trapped inside the stator holes and break up subsequently. Therefore, the size of stator holes and residence time are important factors.

The principle of droplet size reduction is largely attributed to the turbulence effect, which is quite similar to that with HPH but is not as complex as that of rotor-stator homogenizer. The emulsification process is an equilibrium of droplet breakup by disruptive forces and re-coalescence by cohesive forces [58,60]. Firstly, with HPH, the residence time in the valve or nozzle is very short, so that droplet re-coalescence is insignificant. However, the residence time in rotor-stator is much longer and droplet re-coalescence could only be neglected in the case of dilute emulsion with a high surfactant content. Secondly, the dispersed liquid in a rotor-stator homogenizer is more viscous, and the theoretical description of emulsification is more sophisticated. For PCM emulsions, the maximum stable equilibrium droplet size (*d_max_*) can be estimated by [61],
(5)$$d_{max}=A_{1}\left(1+A_{2}\phi \right)N^{-1.2}r^{-0.8}\gamma^{0.6}\rho^{-0.6}$$
where *A_1_* is a constant related to specific geometries of tank and rotor-stator head, *A*_2_ is a constant of the tendency to coalesce, ϕ is the volume fraction of the dispersed phase, *N* is the rotation speed, *r* is the external diameter of rotor, *ρ* is the mass density of the continuous phase.

#### 3.1.4. Ultrasonic Emulsification

Ultrasonic emulsification (UE) is regarded as a more efficient technique for droplet size reduction than rotor-stator homogenizer or HPH [62]. The mechanism of UE is associated with the physical effects of acoustic cavitation such as high-speed micro-jets, high temperature and pressures, shock waves, turbulence and shear forces [63], as well as acoustic streaming [64,65]. Acoustic streaming is formed by the pressure gradient on the axis of the emitter, causing a liquid motion at a velocity of 1–2 m/s [66], which depends on the ultrasound energy density and liquid viscosity. With these effects, the heat, mass and momentum transfer are intensified, promoting the emulsification process.

Acoustic cavitation, a phenomenon of growth and collapse of vapor bubbles in liquid under an ultrasound field, is the primary contributor to UE (Figure 3E). There are two modes of acoustic cavitation behaviors, non-inertial and inertial. Non-inertial acoustic cavitation refers to the symmetrical oscillating of stable vapor bubbles for several acoustic cycles, also called stable cavitation. Inertial acoustic cavitation, on the other hand, is transient in nature, and characterized by the growth and collapse of vapor bubbles. Generally speaking, transient acoustic cavitation plays a major role in the performance of UE.

Higher frequency ultrasound can produce smaller bubbles due to the shorter cycles of compression and rarefaction, leading to a higher temperature and the generation of more free radicals which are suitable for chemical activation [64]. Low-frequency ultrasound, mostly commonly at 20 kHz, gives rise to larger bubbles, which collapse more violently to produce a greater effect for emulsification. Also, the high amplitude ranging 75–100 μm (peak-to-peak) is required to obtain high-quality emulsions with smaller droplet size [67]. Since the maximum amplitude generated by ultrasonic transducers is around 25 μm (peak-to-peak) only, a high-gain acoustic horn is needed to increase the amplitude, which has a small output diameter (10–20 mm). Therefore, UE has been limited to the laboratory-scale applications. Table 3 provides a summary of the characteristics and applications of the different high-energy emulsification techniques.

### 3.2. Studies of PCM Emulsions

As mentioned, instability and supercooling are two major challenges hindering the development and application of advanced PCM emulsions, which have been most widely addressed in previous studies. In this section, the relevant studies are reviewed according to the fabrication methods. So far very few research studies have been documented on PCM emulsions prepared by HPH or ME. Vilasau et al. [68] reported a paraffin emulsion with a melting point of 52 °C with a fixed content of 57.4 wt% prepared by HPH. The average droplet size was 1 μm prepared under a high pressure of 27 MPa and with a surfactant content of 1.8 wt%. However, a dramatic increment in droplet size was observed, reaching to 6.7 μm only after 2 h pipeline circulation test at room temperature, since a low surfactant content could not effectively prevent droplet coalescence after fabrication. Hagelstein and Gschwander [69] also used HPH with pressure in the range of 10–50 MPa to prepare octadecane emulsions using polyvinyl alcohols (PVAs) as the surfactants, resulting in average droplet size as small as 0.4 μm. More importantly, PVAs showed a positive effect on supercooling, decreasing from 12 to 2 °C. This result further suggested that the development and study of such bi-functional materials may be a possible solution to problems of poor stability and supercooling spontaneously.

#### 3.2.1. Formulation by Rotor-Stator Homogenizer

The operating conditions of rotor-stator homogenizer such as rotation speed and temperature are important to determine the droplet size and stability of PCM emulsions. Zhao et al. [70] studied the optimal rotation speed, emulsifying time and temperature for a paraffin wax with a melting point (T_m_) of 59 °C. The emulsifying temperature and rotation speed showed significant impacts on the droplet size. For example, the average droplet size decreased from 9 μm at 70 °C to 0.1 μm at 90 °C and decreased from around 5 μm at 500 rpm to 0.1 μm at 1100 rpm, but increased to over 0.4 μm with a further increase in the rotation speed to 1500 rpm. The increase of emulsification time from 10–30 min, only decreased the size from 1 to 0.1 μm. Fischer et al. [71] also studied conditions for the dispersion system of a paraffin mixture (Crodatherm-47/Crodatherm-53 at 1:1 mass ratio, T_m_ = 50 °C). The shear rate and time were varied with the temperature fixed at 75 °C. When comparing shear rate (20,000 and 50,000 s^−1^) with processing time in 10–60 min, a higher shear rate led to a smaller particle size distribution, and the effect of time in this range was minor.

Zhang et al. [72] and Wang et al. [73] also evaluated the effect of rotation speed on emulsification of paraffin PCM (Figure 5). For relatively large droplet size dispersion, the increase of rotation speed could effectively lower the size from 24 to 4 μm. However, when the particle size had already reached to about 1 μm, a further reduction could be hardly achieved by a higher rotation speed up to 24,000 rpm. The reason has been stated in the above section due to the limit of rotor-stator homogenizer. Under this circumstance, a higher surfactant content is required to obtain a smaller particle size.

Many studies have been devoted to the selection and content of surfactants. Zhang et al. [72] tested several surfactant mixtures of the Tween-Span series at 1:1 mass ratio and found that the Span20/Tween80 combination presented the best performance for octacosane emulsions in terms of the relatively small particle size and breaking ratio. A later study by the same group [74] showed that the Span 80/Tween 80 combination was the best for hexadecane emulsions. Their results indicated that the optimal surfactant selection depended strongly on the paraffin category. The HLB value was also varied when the mass ratio of Span80/Tween80 was adjusted, and the optimal range was identified as 9.5–11, which was also confirmed by Shao et al. [75]. An optimal HLB value of 10.9 was found for commercial paraffin RT10 based on the Brij 52/Span 20 combination. Wang et al. [76] used response surface methodology (RSM) to optimize the HLB value and surfactant content (Tween 80/Span 80) for commercial paraffin OP10E and their results showed that an HLB value of 8.9 and a surfactant content of 5 wt% achieved the smallest droplet size for 30 wt% OP10E emulsion. Based on these previous studies, it can be concluded that the HLB range of 9–11 may be suitable for the majority of paraffin. As for the surfactant content, a higher value was desirable to get the smaller size. Moreover, the oil/surfactant (O/S) mass ratio is relatively high for rotor-stator homogenizer.

Table 4 summarizes the related studies, which follow the order from a low to high melting temperature of paraffin. It is clear that the average droplet size ranged in the micron level, which is relatively large and unfavorable for the stability of paraffin emulsions but favorable for the well-controlled supercooling degree (ΔT). It is easier for the larger droplets to form nuclei, and also for the containment of nucleating agents (NAs). Nanoparticles such as graphite [76], graphene [77], multi-wall carbon nanotubes (MWCNTs) [78] and nano-SiO_2_ [72,74] and organic materials with higher melting points (paraffin [79] and fatty acids [71,80]) have been applied as nucleating agents to lower the supercooling. However, the introduction of NAs usually negatively affects stability. As a result, it is still not feasible to use a rotor-stator homogenizer for the fabrication of PCM emulsions with long-term stability and low supercooling degree.

#### 3.2.2. Formulation by Sonicator

As summarized in Table 4, the paraffin emulsions prepared by UE are mostly nano-emulsions with very small droplets as low as 200 nm. For example, Ho et al. [81,82] prepared eicosane and tricosane nano-emulsions with average droplet size in the range of 150–170 nm using a probe-type sonicator (700 W, 20 kHz). With the sonicator operated at 40% amplitude for 10 h and 20% amplitude for 24 h, 1 kg nano-emulsions of eicosane and tricosane, respectively, were prepared, though the degree of supercooling was as high as 13–14 °C. Zhang and Zhao [83] prepared nano-emulsions of 10–40 wt% octadecane with sodium dodecyl sulphate (SDS) as the emulsifier in an ultrasonic processor (650 W, 20–25 kHz) with small droplets ranging 100–200 nm and but the a large supercooling degree of ~15 °C. To reduce the supercooling, the authors [84] tried to embed nucleating agents, i.e., octadecanol and MWCNTs, into the nano-droplets but only decreased the supercooling to 11 °C. Similarly, in the study of Cabalerio et al. [85] to the nano-emulsions (90–150 nm) of a commercial paraffin RT21HC, the addition of three kinds of paraffin with higher melting temperatures showed a limited effect on supercooling, remaining in the range 7.4–9.5 °C.

More promising results have been reported recently on the reduction of supercooling. Agresti et al. [86] prepared RT55 and RT70HC nano-emulsions by a solvent-assisted route, which involved the ultrasonic homogenization of oil (PCM/solvent) in water fine emulsions, followed by evaporation of the organic solvent (hexane). Since the droplet size increased with the PCM content, 10 wt% was reported as the maximum for maintaining a small size. The addition of RT70HC and single-wall carbon nanohorns (SWCNHs) could effectively suppress the supercooling degree for nano-emulsions of RT50 and RT70HC, respectively. A later work by the same group [87], which still followed the same strategy of adjusting the categories of paraffin and nanoparticles, found that graphene oxide (GO) and reduced graphene oxide (rGO) were ineffective to reduce supercooling for 5 wt% RT21HC nano-emulsion. Although GO could control the supercooling of 5 wt% RT55 nano-emulsion at 1 °C, the supercooling issue was only partially solved since the main freezing peak at around 40 °C still existed, as shown in Figure 6. The inefficient encapsulation of GO within RT55 nanoparticles was a possible cause of the problem.

It appears that the common nucleating agents that are useful to reduce supercooling of conventional PCM emulsions with relatively large droplets are not always effective for the nano-emulsions with much smaller droplets. As the addition of nanoparticles is still regarded as a simple approach toward the supercooling issue, more effort should be made to discover and apply suitable nanoparticles. With an extremely small droplet size and a high dispersibility of paraffin, smaller nanoparticles could be embedded into the nano-emulsions through UE. This may be useful not only to control supercooling but also to enhance stability and thermal conductivity.

As shown in Table 4, UE has been less widely studied than the rotor-stator homogenizer for the development of paraffin nano-emulsions. The selection of surfactants was limited, and the processing parameters were still not systematically optimized. It is envisaged that UE is a promising technique for the fabrication of advanced PCM nano-emulsions with unique functional properties. Therefore, it is worthwhile to put more effort into developing an application of UE in a wide range of PCM emulsions and TES systems.

**Table 4 materials-15-00121-t004:** Summaries of PCM emulsions fabricated by high-energy methods.

PCM	T_m_	Surfactants	O/S Ratio	Size	Conditions	NAs	ΔT	Ref.
**Rotor-Stator Homogenizer**								
Tetradecane	4–6 °C	SDS, Tween 40	-	3–5 μm	24,000 rpm	-	9.7 °C	[79]
Tetradecane	4–6 °C	Triton X/Span60 (1:2)	25:3	~1 μm	5000 rpm (5 min)	-	0 °C	[88]
RT10	10 °C	Brij52/Tween 60 (2:3)	25:3.75	~3 μm	500 rpm (15 min); 7200 rpm (45 min)	-	-	[75]
OP10E	10 °C	Span80/Tween 80 (57:43)	6:1	3–4 μm	8000 rpm (3 min)	Graphite	0 °C	[76]
Hexadecane	18 °C	SDS, Tween 40	-	2–5 μm	24,000 rpm	Paraffin	2.1 °C	[79]
Hexadecane	18 °C	SDS, Tween20, Tween80	-	2.3–4.5 μm	8000 rpm (10 min)	MWCNTs	0 °C	[78]
Hexadecane	18 °C	Span80/Tween80 (1:1)	6:1	0.5–1.2 μm	8000–24,000 rpm (10 min)	SiO_2_	1.2 °C	[74]
RT20	20 °C	SDS, Tween40	-	3–5 μm	24,000 rpm	-	9.2 °C	[79]
RT25HC	25 °C	Ethoxylated alcohols	5:1	>1 μm	20,000 s^−1^ (5 min)	Myristic acid	0 °C	[80]
Paraffin wax	40–44 °C	DAS/Span80	-	10–20 μm	10,000–19,000 rpm (3 min)	-	-	[89]
Paraffin wax	42–44 °C	PEG-600/PVA/SMA with GO nanoparticle	-	~8 μm	19,000 rpm (5 min)	-	-	[90]
Crodatherm−47/53	50 °C	Steareth-2/100 (1:3)	4:1	0.5–5 μm	20,000 s^−1^ (1–60 min) 50,000 s^−1^ (1–150 min)	Boronitride Stearic acid	0 °C	[71]
Paraffin wax	54 °C	PVA/Eumulgin O10 with SiO_2_@Al_2_O_3_	-	0.4–0.5 μm	3000 rpm	-	-	[91]
Paraffin wax	58–60 °C	PVA/PEG-600 (76:24)	20:2.8	4–6 μm	12,000 rpm (5 min)	-	0 °C	[92]
Paraffin wax	58–60 °C	Span80/Tween80 (47:53)	5:2	~0.5 μm	8000 rpm (10 min)	Graphene	0.3 °C	[77]
Paraffin wax	59 °C	Span60/Tween60 (1:3) with resin as stabilizer	2:1	0.1–9 μm	500–1500 rpm (10–45 min)	-	-	[70]
Octacosane	60 °C	Span20/Tween80 (1:1)	4:1	0.5–1.5 μm	8000–24,000 rpm (10 min)	SiO_2_	~1 °C	[72]
Paraffin wax	62–64 °C	PVA/PEG-600 (1:1)	5:1	3–10 μm	6000–14,000 rpm (5 min)	-	0 °C	[73]
**Ultrasonic Emulsification**								
RT21HC	21 °C	SDS	10:0.3	130–160 nm	65 W for 10 min; 95 W for 20 min	GO, rGO	~12 °C	[87]
RT21HC	21 °C	SDS	20:2.5	90–150 nm	65 W (20 kHz)	Paraffin	~9 °C	[85]
Octadecane	28–30 °C	SDS	(10–40):2 10:1	100–200 nm	70% Amplitude for 10 min 70% Amplitude for 30 min (650 W, 20–25 kHz)	Octadecanol MWCNTs	~11 °C	[83,84]
Eicosane Tricosane	36–38 °C 45–47 °C	SLS	4:1	150–170 nm	40% Amplitude for 10 h 20% Amplitude for 24 h (700 W, 20 kHz)	-	~14 °C	[81,82]
RT55	55 °C	SDS	10:0.3	200 nm	65 W for 10 min; 95 W for 20 min	GO, rGO	1 °C	[87]
RT55	55 °C	SDS	8:1	173 nm	65 W for 10 min (20 kHz)	Paraffin	2.2 °C	[86]
Paraffin wax	58–60 °C	Pluronics P-123	20:1	~600 nm	20% Amplitude for 60 min (700 W, 20 kHz)	-	3–4 °C	[93]
RT70HC	70 °C	SDS	8:1	166 nm	65 W for 10 min (20 kHz)	SWCNHs	1.8 °C	[86]

## 4. Preparation of PCM Emulsions by Low-Energy Methods

### 4.1. The Basic Mechanisms

The emulsification with low-energy methods uses the internal chemical energy of the dispersion system to form fine emulsions, and the energy consumption is 3–5 orders lower than that with the high-energy methods. The low-energy methods with no changes in the spontaneous surfactant curvature, kwown as self-emulsification (SE), generally involve two steps, firstly the mixing of organic phase and hydrophilic surfactant, and the titration of organic phase into aqueous phase at a constant temperature and a low stirring speed. The proposed mechanism could be related to the surfactant movement that results in the formation of a bicontinuous microemulsion at the boundary, which breaks up, leading to the spontaneous generation of fine oil droplets [23]. A certain range of surfactant-oil-water ratios determines the bicontinuous microemulsion formation; several processing parameters such as operating temperature, stirring speed, and rate of addition affect its breakdown.

When the surfactant spontaneous curvature change is triggered by a change in temperature or composition, the method is designated as phase inversion temperature (PIT) or phase inversion composition (PIC). The experimental set-up with the PIC method is the opposite of SE, which includes the mixing of organic phase and surfactants, and titration of the aqueous phase into the organic phase at constant temperature and low, stirring speed. The PIT method is introduced by Shinoda [94] based on temperature-sensitive surfactants, e.g., nonionic ethoxylated-type that could be denoted as C_i_E_j_, which involves three major steps are as follows: (i) the formation of coarse emulsion at room temperature; (ii) heating of the coarse emulsion to around or above the PIT temperature; and (iii) fast cooling to room temperature at low stirring speed. Phase behavior of the surfactants plays an essential role in the properties of emulsions. Phase inversion crossing lamellar liquid crystalline phase (Lα) or bicontinuous microemulsion phase (D) is responsible for the forming of nano-droplets by the PIT or PIC methods [95].

The mechanism and Lα or D phase formation can be briefly described by the critical packing parameter (CPP=V/la) [96] of ethoxylated-type surfactants, where *V* is the volume of hydrophobic group, *l* is the length of hydrocarbon chain, and *a* is the cross-sectional area of hydrophilic group. The interaction of hydrophilic ethylene oxide (EO) groups in ethoxylated-type surfactants with water is strongly affected by temperature and water content. At a higher temperature or a lower water content, the hydration effect of EO groups is weak with a shrinkage area leading to a high CPP value (>1). The surfactant is more soluble in oil phase and the surfactant curvature is negative, thereby favoring the formation of W/O emulsions. With the decrease of temperature or addition of water, the interaction of EO groups with water is gradually stronger. When CPP ≈ 1, Lα or D phases are formed as a balance of hydrophilic and hydrophobic surfactant with zero curvature. This temperature is referred to as the PIT point or HLB temperature of surfactant, where the interfacial tension is extremely low. Therefore, nano-droplets could be formed more easily by crossing this region. On the other hand, the produced O/W nano-emulsions (CPP < 1) should be immediately move away from the phase inversion region or the emulsions may break down quickly by coalescence. In this aspect, PIT is better than PIC for producing smaller droplets with a uniform size distribution [97], since the rate of temperature decrease is usually high in the PIT method, while that rate of water addition is relatively low in the PIC method. Overall, both PIT and PIC are surfactant HLB-balanced methods.

However, the HLB value is not enough to describe the balanced state. Therefore, the concept of Hydrophilic−Lipophilic Deviation (HLD) has been developed [98]. In addition, the PIT point assessment of various ethoxylated-type nonionic surfactants has been conducted based on a C_10_E_4_/octane/water system [99]. In the experiment, a second surfactant was added, and the PIT point variation as a function of the concentration (dPIT/dC) was analyzed and summarized in Figure 7. The negative value indicated that the added surfactant caused a reduced PIT point, and therefore it was less hydrophilic than C_10_E_4_. Compared with the HLB value, the value of dPIT/dC would be more instructive for surfactant selection with proper PIT points.

### 4.2. Preparation and Characterization of PCM Emulsions

#### 4.2.1. PIC Method

Hessien et al. [100] studied the effects of emulsifying temperature and O/S mass ratio with the PIC method on the average droplet size of nano-emulsions. Binary surfactants, Span80/Tween80 with an HLB value of 11.6, were mixed with paraffin oil, and then water was added dropwise at 2 mL/min under a slow stirring speed of 60 rpm. With the high emulsifying temperature (70–80 °C), the 2:1 O/S mass ratio attained the smallest droplet size of~100 nm; while the higher or lower O/S mass ratio increased the droplet size to 150–170 nm. A lower emulsifying temperature would also lead to bigger droplet sizes to over 150 nm at 2:1 O/S mass ratio.

Kim and Cho [101] also reported a similar effect of emulsifying temperature by the PIC method. At O/S mass ratio of 1:1, nano-emulsion of paraffin oil showed a gradual decrease in the droplet size from 120 to 40 nm with a temperature from 30–80 °C. At an emulsifying temperature of 70 °C, when the HLB value of the surfactant mixture of Span80/Tween80 was over 12, the droplet size remained almost unchanged at about 50 nm. Other studies from the group [102,103] showed that the higher emulsifying temperature enabled paraffin emulsions to form smaller droplets with higher stability, but the optimal HLB value of Span80/Tween80 was slightly different, 9.5–10.3 and 10–11 for paraffin wax and paraffin oil, respectively (Figure 8 and Table 5). Therefore, the general trend is that the higher emulsifying temperature is desirable to fabricate nano-sized PCM emulsions, while the optimal HLB value of the combined Span80/Tween80 depends on the categories of paraffin.

The stability modification of paraffin oil emulsions with Span20/Tween20 by adding extra ionic surfactants or polymers after emulsification has also been evaluated [104,105]. For ionic surfactants, both SDS and cetyltrimethylammonium bromide (CTAB) could reduce the droplet size. The zeta potential was changed (enhanced or weakened) by adding SDS or CTAB, implying that the higher stability of nano-emulsion with SDS is attributed to an increased electrostatic interaction.

In addition to paraffin, fatty acids, i.e., stearic acid and myristic acid, as the PCMs have been emulsified by the PIC method [106]. Although the droplet sizes of nano-emulsions were all below 80 nm, stearic acid outperformed myristic acid under the same conditions (3:2 O/S mass ratio and 1:1 SDS/Span85 mass ratio). One of the possible reasons was related to the neutralized fatty acid surfactants formed by slowly adding NaOH solution. Table 5 summarizes the studies of PCM emulsions prepared by low-energy methods. It is noticed that the supercooling degree for emulsions prepared by the PIC method was rarely characterized.

**Table 5 materials-15-00121-t005:** Summaries of PCM emulsions fabricated by low-energy methods.

PCM	T_m_	Surfactants	O/S Ratio	Size	Methods	NAs	ΔT	Ref.
Paraffin oil (ρ = 0.83 g/cm^3^)	-	Span80/Tween80 (HLB = 12–13)	1:1	~50 nm	PIC	-	-	[101]
Paraffin oil (ρ = 0.85 g/cm^3^)	-	Span20/Tween20 (HLB = 12.75) with ionic surfactants or polymers	2:1	80–200 nm	PIC	-	-	[104,105]
Paraffin oil (ρ = 0.85 g/cm^3^)	-	Span80/Tween80 (HLB = 10.3) with CTAB and NaBr	-	100–200 nm	PIT	-	-	[107]
Paraffin oil (ρ = 0.86 g/cm^3^)	-	Span80/Tween80 (HLB = 11.6)	Varies	95–170 nm	PIC	-	-	[100]
Paraffin oil (ρ = 0.86 g/cm^3^)	-	Span80/Tween80 (HLB = 10–11)	1:1	~50 nm	PIC	-	-	[103]
Tetradecane	4–6 °C	Span60/Tween60 (1:2)	10:3	200–220 nm	PIT/PIC	-	~14 °C	[108]
Tetradecane	4–6 °C	Span80/Tween80 (HLB = 11.7)	5:2	~500 nm	UE/PIT	Polymer	~3 °C	[109]
Tetradecane	4–6 °C	Self-synthesized POP surfactants	-	40–90 nm	PIT	-	-	[110]
RT10	10 °C	Span60/Tween60 (1:2)	10:3	157 nm	PIT	-	~4 °C	[111]
Parafol 15-90	15 °C	Span60/Tween60 (1:2)	10:3	145 nm	PIT	-	~12 °C	[111]
Hexadecane	18	Span60/Tween60 (1:2)	10:3	145 nm	PIT	-	~11 °C	[111]
Hexadecane	18 °C	Brij L4	30:11 30:13	60 nm 126 nm	PIT PIC	- SiO_2_	~9℃ ~8 °C	[97]
Hexadecane	18 °C	Brij L4/Tween60 (3:2) Brij L4/Polymer (4:1)	20:11	~60 nm	PIT	Paraffin -	2–3 °C 5 °C	[112]
Hexadecane	18 °C	Tween 60, Tween 80, Brij O10	-	422 nm	D-phase	-	~17 °C	[113]
Hexadecane	18℃	Tween 80	5:2	223 nm	D-phase	-	-	[114]
Hexadecane	18 °C	Span80/Tween80 (HLB = 12)	5:2	290 nm	D-phase		~15 °C	[115]
Octadecane	28–30 °C	Tween 80	5:2	220–240 nm	D-phase	-	-	[114]
Octadecane	28–30 °C	Span80/Tween80 (HLB = 12)	5:2	320 nm	D-phase	-	~13 °C	[115]
Stearic acid Myristic acid	53 °C	SDS/Span85 (1:1)	3:2	40–80 nm	PIC	-	-	[106]
Paraffin wax	51–54 °C	Span80/Tween80 (HLB = 9.5–10.3)	4:1	600–900 nm	PIC	-	~15 °C	[102]

### 4.2.2. PIT Method

Zhang et al. [97] compared PIT and PIC methods for the preparation of hexadecane emulsions with single Brij L4 as surfactant, and showed that the droplet size of PCM emulsions by the PIT method was smaller than that by the PIC method (Figure 9). The PIT method was less sensitive to the variation of surfactant content, but the minimum droplet size was obtained at the medium value for both the PIT and PIC methods. The higher surfactant content would cause an increase in the viscosity of emulsions, which may affect the phase inversion process and result in larger droplet size. In addition, the PIT point was gradually reduced by the increasing surfactant content. The supercooling degree was quite high at about 9 °C and the addition of nano-SiO_2_ as NA had little effect. In addition, it might be more difficult for the low-energy methods than the high-energy methods (e.g., UE) to encapsulate nanoparticles within nano-droplet. Hence, organic NAs or polymers should be more suitable for low-energy methods.

In a recent study from our group [112], combined surfactants (Brij L4 with Tween60 or polyethylene-block-poly(ethylene glycol), PE-*b*-PEG) were used to fabricate hexadecane nano-emulsion by the PIT method. The addition of a di-block polymer not only enhanced the stability, but also reduced the supercooling degree from 9 to 5 °C, functioning like PVAs as previously discussed. The use of hydrophilic surfactant Tween60 could elevate the PIT point, and the synergistic effect of mixed surfactants allowed the nano-emulsions to remain stable in a wider temperature range. Paraffin with a higher melting point was added as a NA, resulting in a low supercooling degree of 2.4 °C. However, the problem was only partially resolved due to the second freezing peak in a lower temperature range.

Schalbart et al. [108] comprehensively investigated multiple effects of surfactant location, emulsification temperature, and cooling rate based on tetradecane emulsions (10:3 O/S mass ratio; 1:2 Span60/Tween60 mass ratio). Surfactant location of Span 60 in oil and Tween 60 in water and low cooling rate by air prohibited the formation of stable tetradecane emulsions. Comparatively, the best emulsification conditions were identified by mixing all components (coarse emulsion) at room temperature, and then heating the mixture over the PIT point, followed by immediately adding cold water. In fact, this procedure combined the PIT and PIC method, and the smallest droplet size of 206 nm was attained. Schalbart et al. [111] further explored the differences in PIT point and properties of emulsions prepared with different PCM paraffin (tetradecane, hexadecane, RT10 and Parafol 15–90). The PIT point of commercial paraffin Parafol 15–90 was 59 °C, which was lower than those of three other paraffin (72–74 °C). At 10:3 O/S mass ratio, while the droplet size was similar at around 150 nm with different paraffin. The supercooling degree was lowest about 4 °C with the RT10 nano-emulsion and was over 10 °C with other nano-emulsions.

Fumoto et al. [109] combined the UE and PIT methods to fabricate tetradecane emulsion that was stabilized by a mixture of Span80/Tween80 (HLB = 11.7) at 5:2 O/S mass ratio. The PIT point was estimated with an equation but not determined by measurement, which might not be so accurate for controlling emulsion condition. As a result, the average droplet size was ~500 nm. In addition, a polymer (thermoplastic vulcanizate) was used as NA, reducing the supercooling degree from 13.5 to 2.9 °C. Ren et al. [110] synthesized temperature-sensitive surfactants with short polyoxypropylene (POP) chains (2.5–6 units), which could be used to prepare nano-emulsions by the PIT method. The PIT point was slightly increased, from 53 to 56.5 °C, by increasing the POP surfactant content (4.8–7.9 wt%), and the droplet size gradually decreased from 90 to ~40 nm.

### 4.2.3. D-Phase Method

The D-phase method is another low-energy method developed by Sagitani et al. [116] and but has been less studied for PCM emulsions. The procedure is composed of three steps as follows: (i) mixing polyalcohol, water and nonionic surfactants as the D-phase, (ii) adding oil to form O/D gel and (iii) water dilution with stirring to form fine emulsions. A special advantage of this method is there is no need to adjust the HLB value of the surfactant. Kawanami et al. [113] first used this method to prepare PCM emulsions with various surfactants and surfactant mass ratios to water. The final hexadecane emulsion had an average droplet size of 422 nm but a very high supercooling, up to 17 °C.

Morimoto et al. [114] further modified the recipe of the O/D gel (1:1:2:5 of water/1,3-butanediol/Tween80/PCM), and attained smaller droplets of 220–240 nm of hexadecane and octadecane emulsions. The mixed surfactant (Span80/Tween80 at 7:18 mass ratio) was further explored based on the above formula of the O/D gel by Chen and Zhang [115]. Although the droplet size of hexadecane emulsion was slightly larger than that of the emulsion prepared with only Tween 80, the stability was high (7 months vs. 2 months). The stirring effect was also evaluated, and the results showed that stronger mixing was more favorable. Similarly, the supercooling degree was high due to the small droplet size and lack of a NA.

Compared with high-energy methods, the low-energy methods can achieve a droplet size as small as <50 nm, but require a higher surfactant content. Supercooling problem is quite severe and not effectively controlled by the addition of NAs. Firstly, the addition of NAs may affect the emulsion stability or even cause the failure of emulsion formation. Secondly, it is hard to encapsulate the NAs such as nanoparticles within the nano-droplets of paraffin due to inefficient energy input. Overall, considerable effort is needed to overcome the drawbacks of the low-energy methods to be applied to the preparation of favorable paraffin emulsions.

## 5. Applications

Because of the low melting point range of paraffin and the poor stability of paraffin emulsions, especially in the high-temperature range (>60 °C), their potential applications may include building HVAC systems and thermal management systems for photovoltaic (PV) modules or Li-Ion batteries. The building sector is a major energy consumer, accounting for more than 40% of the overall energy generated globally [117], particularly in the HVAC systems, which account for a large proportion. The development and utilization of TES systems for cooling and heating may balance the energy demand between peak and off-peak period and reduce the usage of electricity. Such a TES system usually has a storage tank for energy storage and a heat exchange unit for energy release. It is noticed that most studies have focused on the storage tank for the characterization of thermal storage capacity and heat transfer rate of PCM emulsions. As for the heat exchange unit, cooling/heating performance for energy saving in buildings has rarely been studied.

Delgado et al. [118] experimentally studied a PCM emulsion in the storage tank. The low-cost paraffin as the PCM was derived from the petroleum refining process, and PCM emulsions with an average droplet size of 1 μm had a phase change temperature range in 30–50 °C, which was confined in a 46 L storage tank (Figure 10). The overall heat transfer coefficient of PCM emulsion was 2–6 times higher than that of the solid–liquid PCMs in conventional LHS systems but five times lower than for the tank filling with water. Fortunately, the thermal storage capacity of the emulsion tank was around 34% higher than that of water tank. Due to the low heat transfer rate, a stirrer was then instilled in the upper part at the central axis of the storage tank [119]. With agitation, the overall heat transfer coefficient was significantly enhanced, around 5.5 times higher, but still lower for water tank with the same stirring speed. The reason was most likely associated with the high PCM content (60 wt%) and large viscosity, 2–5 orders magnitude higher than that of water. As a consequence, PCM emulsion was not suitable for pipe transportation but confined in the storage tank as a stationary LHS unit.

Then, a lower viscous 20 wt% n-hexadecane nano-emulsion (~80 nm) was developed and evaluated in a lab-scale TES system (Figure 11) [120]. The PCM nano-emulsion working as HTF and flowing in pipes had a phase change temperature range in 10–15 °C. The cooling energy was firstly stored in the nano-emulsion during charging. Compared with the above analyzed LHS tank, it had a faster heat transfer rate or charging rate, which was even higher if water flowed in the system. Although energy release through ceiling panels was also conducted in discharging, it was hard to analysis the cooling impact on the room due to their unmatched sizes. A much bigger storage tank (0.5 m^3^) was studied by Biedenbach et al. [121]. The circulation of 600 L n-octadecane emulsion (35 wt%) was performed between the tank and the chiller for cooling energy storage. Similar results, such as the higher thermal storage capacity, 2.6-fold higher than water tank in the temperature range of 20–27 °C, were obtained, but the cooling performance for the room was not conducted in this study.

Li-Ion batteries or PV modules have shown tremendous success in the market for storage or production of electricity, which play an important role to the reduction of carbon emission. Typically, the operating temperature of Li-Ion batteries would affect the overall electrical performance. At higher temperatures, round-trip efficiency decreases, compromising the reliability, cycle life, output power, and energy capacity [122]. Similarly, with the PV modules, the efficiency of electricity generation falls by 0.5% for every 1 °C increment for Si-based solar cells [123]. Therefore, the development of cooling systems based on PCM emulsions for PV panels and Li-Ion batteries is meaningful and crucial.

For Li-Ion batteries, wang et al. [124] studied the cooling performance of 10 wt% OP28E emulsions as the cooling fluid for a cylindrical battery pack. The maximum temperature (T_max_) and maximum temperature distribution (ΔT_max_) of the battery with the emulsion were 1.1 °C and 0.8 °C, respectively, which were lower than those with water at a flow rate of 12 L/h and a discharge rate of 2C. Although an increasing PCM content to 20 wt% could further reduce the T_max_ and ΔT_max_, the pressure drop was also higher. Therefore, 10 wt% was recommended as the optimal mass fraction.

Later, this research group [125] further studied the cooling performance of 10 wt% paraffin emulsions with higher melting points (OP35E and OP44E) for a pouch battery pack. At a high discharge rate of 9C and fixed flow rate of 10 L/h, T_max_ and ΔT_max_ varied from the inlet temperature of cooling fluid as shown in Figure 12. Specifically, at inlet temperature 30 °C, OP35E emulsions had the smallest T_max_ and ΔT_max_, implying the best cooling performance since PCM melted when flowing from the battery. Similar is the situation for OP44E emulsions at inlet temperature of 40 °C. If there was less PCM melted (at inlet temperature 35 °C), the cooling performance of water and paraffin emulsions had no difference. Therefore, the selection of PCM melting range and inlet temperature depends strongly on the specific system design. Later, Cao et al. [126] further evaluated the cooling performance of OP44E emulsion for battery that was surrounded by composite PCM, achieving enhanced cooling efficiency with less pumping power consumption.

As for PV modules, Feng et al. [127] built a cooling system for a PV panel under indoor conditions using OP35E emulsions and water as cooling fluids. As shown in Figure 13, the PCM content had a limited effect on the temperature reduction, and hence 10 wt% was still regarded as the proper PCM content. The emulsion with a high supercooling degree showed worse performance than water, which suggested that supercooling degree was required to be well addressed to attain a higher electrical efficiency. In terms of PCM mass fraction, 10% was also recognized as a suitable value in some publications using MPCM slurries as HTF in thermal management system for PV modules [128,129]. Here it must be noted that all these previous studies were based on lab-scale systems. Due to a very small energy demand, a low flow rate of emulsion with a low PCM content was suitable. If the energy demand was higher, at least a few kilowatts, a higher PCM content at a relatively larger velocity would be more advantageous in comparison with water in terms of cooling performance and pumping power consumption. On the other hand, the maximum 30% PCM mass friction is recommended if considering the pumping transportation in pipes.

## 6. Conclusions

Through this review, two major issues have been discussed for the development of PCM emulsions from the perspective of fabrication methods. The supercooling issue could be well addressed in PCM emulsions prepared by rotor-stator homogenizer, though the stability is usually poor due to the relatively large droplet size (>1 μm). Low-energy methods could be applied to fabricate nano-sized emulsions with excellent stability, but the supercooling is still not well under control due to the difficulty in the encapsulation of effective nucleating agents such as nanoparticles or the less effectiveness of organic nucleating agents (e.g., paraffin). Therefore, for laboratory studies, ultrasonic emulsification has been regarded as the most promising technique which is possible to produce paraffin nano-emulsions with negligible supercooling degree. However, if considering their future applications, high-pressure homogenizer or microfluidizer should be drawn more attention, since these techniques are more favorable than ultrasonic devices for large-scale production. Although low-energy methods such as the phase inversion temperature method are also feasible industrially, the scale-up faces more challenges. PCM emulsions have shown the potential merits of higher heat capacity and low pumping power, which could be advantageous in many research fields, though case studies are still scarce (one-digit level since 2015). The lack of advanced PCM emulsions with easy scale-up production measures is the key point. Overall, the following areas are highly recommended for future studies of PCM emulsions.

(1)As the effective nucleation agent for emulsions/nano-emulsions, novel nanoparticles with smaller sizes (e.g., quantum dots) or modified interfacial properties could be one of the most promising solutions for effective reduction of supercooling in both high-energy and low-energy methods, which would also bring extra benefits such as enhanced stability and thermal conductivity.(2)Block copolymers or comb-like polymers are meaningful to be screened, which could work as surfactants and nucleating agents like PE-*b*-PEG.(3)The rarely investigated high-pressure homogenizer or microfluidizer and less investigated probe-type sonicator should be conducted, and effects of their processing parameters on properties of PCM emulsions need to be evaluated comprehensively.(4)Due to the instability characteristic of PCM emulsions, the concept of life cycle assessment should be established and evaluated by continuously monitoring property changes of PCM emulsions against operation time in lab-scale case studies, which would offer useful guidance in future large-scale systems.(5)The application fields should be expanded, such as for electronics thermal management. In addition, the current practical applications should be enriched such as cooling performance evaluation for PV modules under outdoor conditions and cooling/heating performance to the room through well-designed heat exchange units.

## Figures and Tables

**Figure 1 materials-15-00121-f001:**
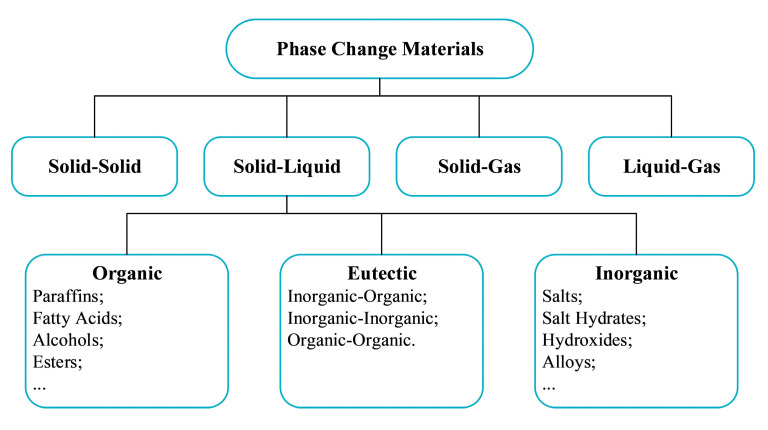
Classification of PCMs.

**Figure 2 materials-15-00121-f002:**
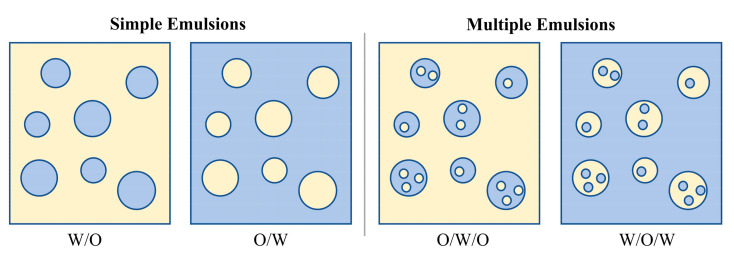
The four basic forms of simple and multiple emulsions.

**Figure 3 materials-15-00121-f003:**
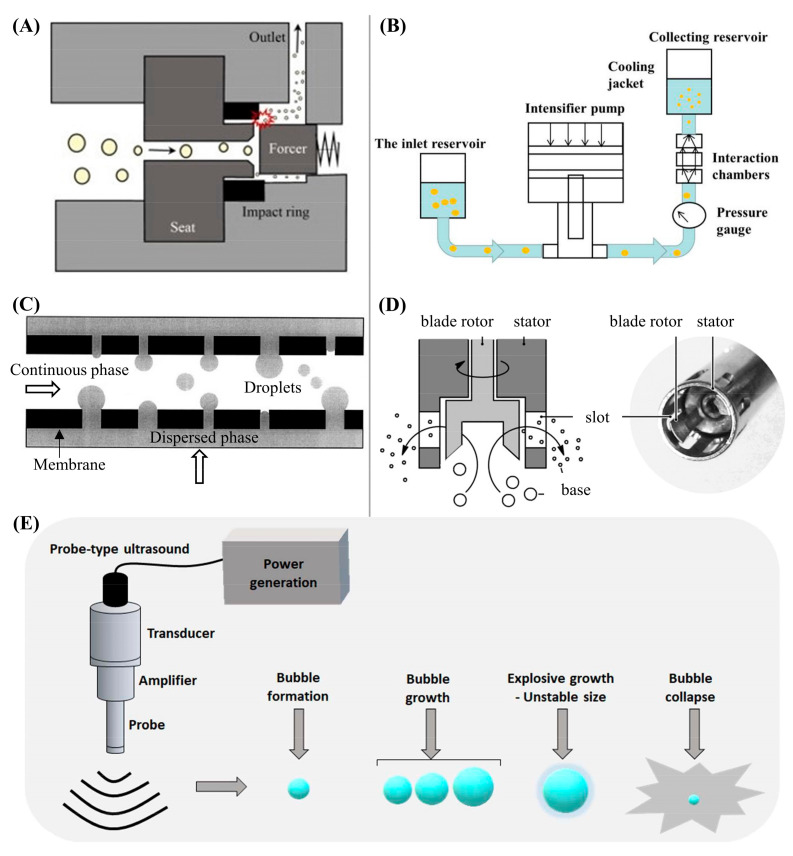
Illustrations of high-energy methods. (**A**) High-pressure homogenization (Reproduced with permission from Ref. [38]. Copyright 2016 Elsevier); (**B**) High-pressure microfluidization Reproduced with permission from Ref. [39]. Copyright 2020 Elsevier); (**C**) Membrane emulsification [40]; (**D**) Rotor-stator homogenization (Reproduced with permission from Ref. [41]. Copyright 2016 John Wiley and Sons); (**E**) Ultrasonic emulsification (Reproduced with permission from Ref. [42]. Copyright 2020 Elsevier).

**Figure 4 materials-15-00121-f004:**
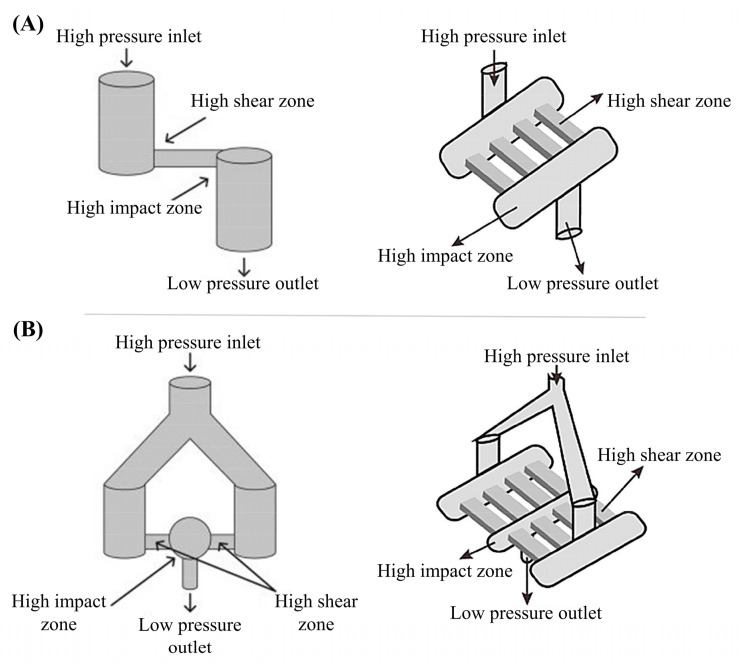
Two geometries of interaction chamber in high-pressure microfluidizer. (**A**) Z-type with single and multiple channels; (**B**) Y-type with single and multiple channels (Reproduced with permission from Refs. [50,51]. Copyright 2018 and 2016 Elsevier).

**Figure 5 materials-15-00121-f005:**
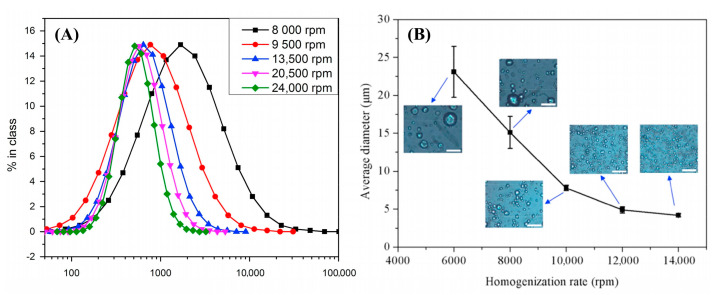
Variation of rotation speed on emulsions. (**A**) 20% octacosane and 5% surfactant mixture of Tween80 and Span20 at 1:1 mass ratio (Reproduced with permission from Ref. [72], Copyright 2016 Elsevier); (**B**) 20% paraffin and 4% surfactant mixture of PVA and PEG-600 at 1:1 mass ratio (Reproduced with permission from Ref. [73]. Copyright 2018 Elsevier) (All % valves in weight %).

**Figure 6 materials-15-00121-f006:**
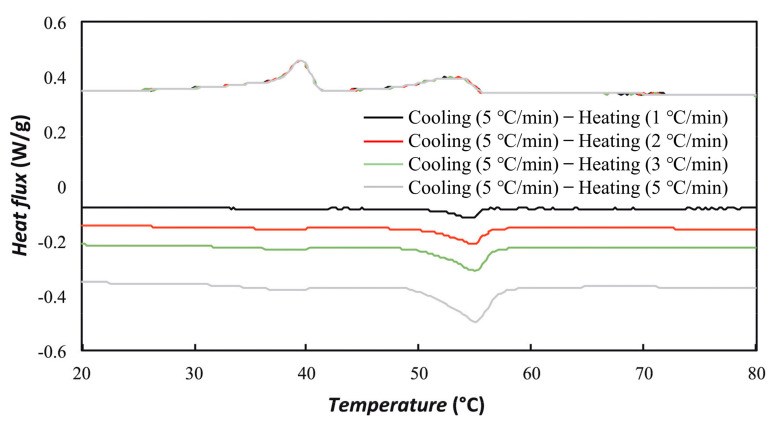
DSC curves of 5 wt% RT55 nano-emulsions with GO as nucleating agent (Reproduced with permission from Ref. [87]. Copyright 2021 Elsevier).

**Figure 7 materials-15-00121-f007:**
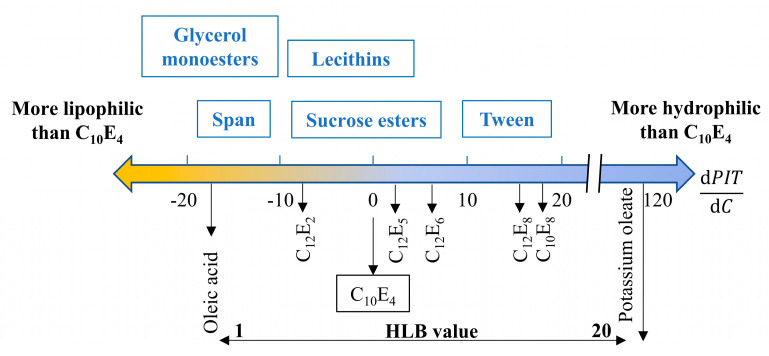
Summary of PIT point variation by adding a second surfactant in a C_10_E_4_/Octane/Water system (Reproduced with permission from Ref. [99]. Copyright 2014 Elsevier).

**Figure 8 materials-15-00121-f008:**
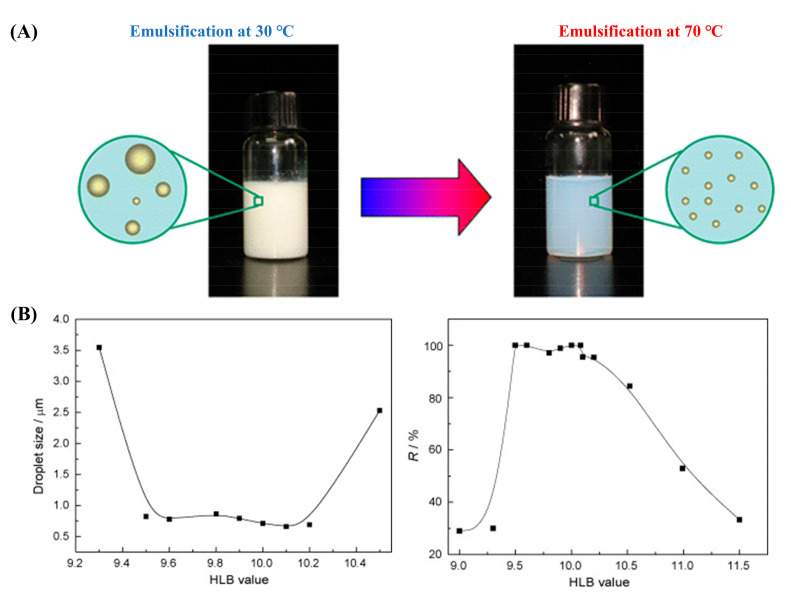
Effects of emulsifying temperature and HLB values (Span80/Tween80) on droplet size and stability of paraffin emulsions prepared by the PIC method: paraffin wax (**A**) and paraffin oil (**B**) (Reproduced with permission from Refs. [102,103]. Copyright 2010 Elsevier and Copyright 2012 American Chemical Society) (*R*% equals the emulsion volume divided by the total volume).

**Figure 9 materials-15-00121-f009:**
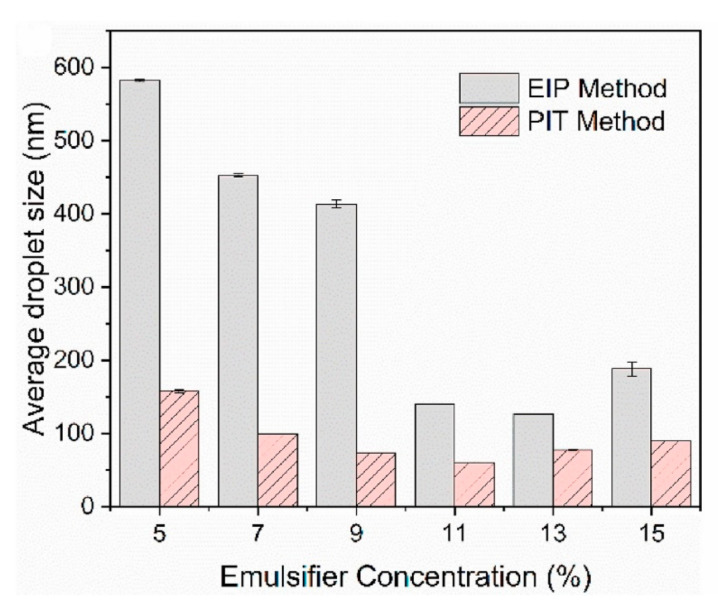
Comparison of droplet sizes of hexadecane emulsion between PIT and PIC methods (Reproduced with permission from Ref. [97]. Copyright 2021 Elsevier).

**Figure 10 materials-15-00121-f010:**
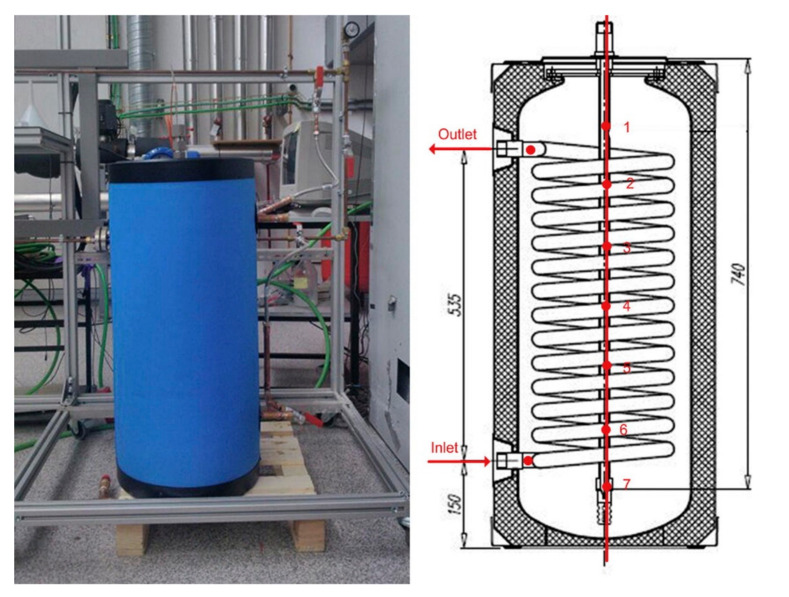
The photograph and geometry of 46 L storage tank (Reproduced with permission from Ref. [118]. Copyright 2015 Elsevier).

**Figure 11 materials-15-00121-f011:**
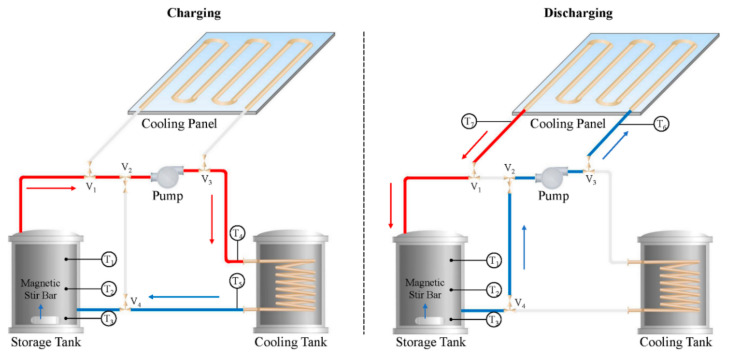
Schematic of charging and discharging mode in a lab-scale TES system (Reproduced with permission from Ref. [120]. Copyright 2021 Elsevier).

**Figure 12 materials-15-00121-f012:**
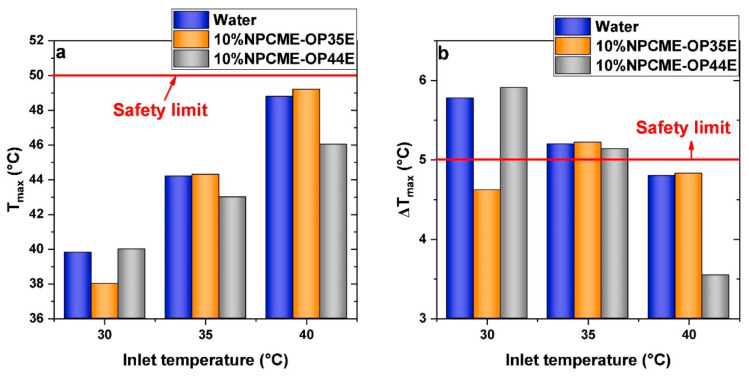
Comparison of T_max_ (**a**) and ΔT_max_ (**b**) with different inlet temperatures of cooling fluid at 9C discharge and 10 L/h flow rate (Reproduced with permission from Ref. [125]. Copyright 2020 Elsevier).

**Figure 13 materials-15-00121-f013:**
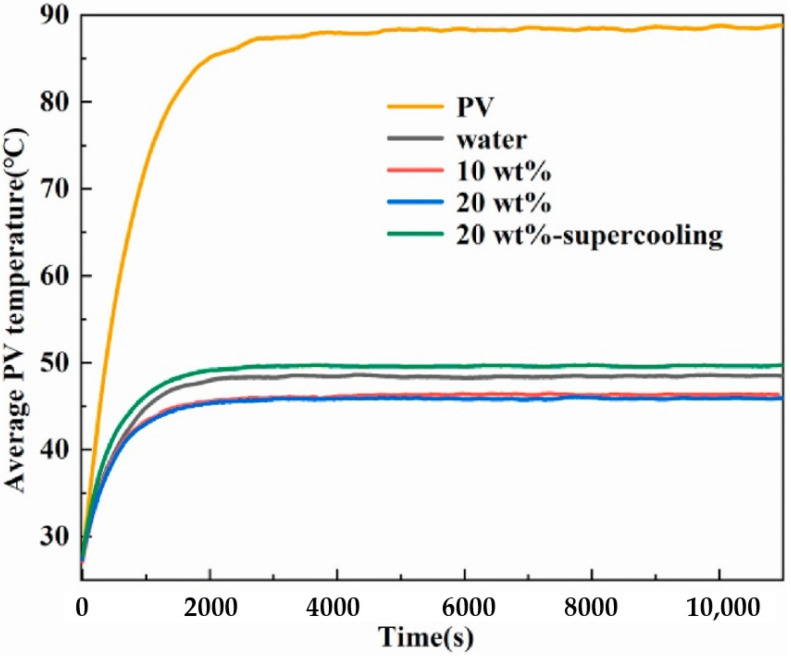
Average surface temperature of PV module at a flow rate of 10 L/h and a constant inlet temperature of 30 °C (Reproduced with permission from Ref. [127]. Copyright 2021 Elsevier).

**Table 1 materials-15-00121-t001:** List of review articles on PCM emulsions in the last 10 years.

Topic	Ref.
PCM emulsions and microencapsulated PCM material slurries: materials, heat transfer studies and applications	Delgado et al. (2012) [8]
State of the art on PCM slurries	Youssef et al. (2013) [9]
PCM emulsions and their applications in HVAC systems	Shao et al. (2015) [10]
A comprehensive review on PCM emulsions: fabrication, characteristics, and heat transfer performance	Wang et al. (2019) [11]
PCM dispersions: thermo-rheological performance for cooling applications	O’Neil et al. (2021) [12]

**Table 2 materials-15-00121-t002:** Comparison of macro-, nano- and micro-emulsions.

Characteristics	Emulsion	Nano-Emulsion	Micro-Emulsion
Size	>200 nm	20–200 nm	5–50 nm
Shape	Spherical	Spherical	Spherical, lamellar
Stability	Thermodynamically and kinetically unstable	Thermodynamically unstable; kinetically stable	Thermodynamically and kinetically stable
Appearance	Milky white	Transparent, translucent, or creamy;	Transparent
Fabrication	High-energy methods Low-energy methods	High-energy methods Low-energy methods	Low-energy methods

**Table 3 materials-15-00121-t003:** Comparison of different high-energy methods.

Conditions	HPH	ME	HSP	UE
Common devices	Microfluidizer; valves	SPG/PTFE membranes	Rotor-stator homogenizer	Probe-type ultrasonicator
Droplet breakup mechanism	Turbulence; shear stress; cavitation	Drop-by-drop	Turbulence; shear stress	Cavitation
Minimum size	0.1 μm	0.2–0.5 μm	1 μm	0.1 μm
Residence time	0.1 < t < 3 ms	-	0.1 < t < 1 s	-
Viscosity range	Low-medium	Low-medium	Low-high	Low
Energy density	Medium-high	Low-medium	Low-high	Medium-high
Energy efficiency	High	Very high	Low	High
Cost	Medium-high	Medium-high	Low	Low-medium
Application	Lab/industrial	Lab	Lab/industrial	Lab

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
