# Peer review of "Preparation of Stable Phase Change Material Emulsions for Thermal Energy Storage and Thermal Management Applications: A Review"

_materials, 2021, doi:10.3390/ma15010121_

Round 1

Reviewer 1 Report

In this review, the supercooling degree and PCM particle size of PCM emulsions fabricated from various methods, surfactants and nucleation agents were summarized. This review is well-prepared and well-written. This review can be considered for publication after Minor revisions summarized as follows:

  1. In the abstract (line 16–18), the authors stated that the small droplet size of PCM emulsions is favorable for higher heat transfer rate. However, the reviewer think that heat transfer rate is not always improved when the particle size becomes smaller. Nomura et al [1]. reported that the heat transfer was promoted when the particle size with respect to the size of the flow path got large. The sentence should be carefully investigated.

  1. In line 154–155, the term “thermodynamically unstable” is ambiguous. The reviewer think it is better to state specifically why unstable it is.

  1. In table 4, it seems that the reference number does not match the content.

[1] K. Nomura and H. Kumano, Int. J. Heat Mass Transf., 116(2018), 1026–1035.

Reviewer 2 Report

The authors presented a review work on "Preparation of Stable Phase Change Material Emulsions for Thermal Energy Storage Applications". The topic is of much interest among the research community and very limited reviews are available on PCM emulsions. I have some serious comments on this work and should be addressed before considering further.

  1. Abstract need to revised in a way that it should project the necessity and content of the review work.
  2. Introduction looks very brief. It can be elaborated with importance and applications of PCM emulsions.
  3. The order of sections needs modification. Section 2.3 can be moved back and can be presented along with stability. Because, these problems exist in PCM emulsions prepared through high and low energy methods.
  4. Section 3.1.4 and 3.2.2 is same title. Section 3.2 can be moved back and presented with more appropriate title.
  5. Restructure the manuscript properly.
  6. Application section can be improved with more discussions.
  7. Conclusion is vague. Must be improved.
  8. The language of this manuscript should be checked.

Reviewer 3 Report

Authors conducted a literature survey on preparation of stable PCM emulsions for thermal energy storage application. The paper is well-prepared and presented. However, the following points should be addressed before it is considered for publication:

  • In the end of the Introduction, the aim of the study is mentioned. This part should be improved referring to similar works in the field and which gap is identified.
  • The methodology followed for the literature survey should be presented. For instance, how are the studied papers selected?
  • It is suggested to enrich the literature survey such as Comprehensive analysis of preparation strategies for phase change nanocomposites and nanofluids with brief overview of safety equipment.
  • Authors might comment on the cost of the emulsion techniques.
  • The title of the study includes "... thermal energy storage applications" , Thus, I suggest expanding Section 5 Applications.
  • Authors discuss the utilization of PCM for "lithium-ion batteries" in Section 5. This seems in contradiction with the title "thermal energy storage applications". Is the purpose in li-ion batteries thermal management or thermal energy storage?
  • Future works suggested in Conclusion section are very helpful to newcomers. However, the findings summarized in the conclusion section should be supported by quantitative data.

My overall opinion about the manuscript is that it is eligible for publication after fixing above given issues. 

Round 2

Reviewer 2 Report

The authors considered all the reviewer comments and revised the manuscript. Now it may be considered for publication.